# Uncertainty quantification and sensitivity analysis of neuron models with ion concentration dynamics

**Letizia Signorelli**[1,3]*, **Andrea Manzoni**[2], **Marte J. Sætra**[3]

**1** Department of Mathematics, Politecnico di Milano, Milano, Italy, **2** MOX, Department of Mathematics, Politecnico di Milano, Milano, Italy, **3** Department of Numerical Analysis and Scientific Computing, Simula Research Laboratory, Oslo, Norway

* letizia.signorelli@mail.polimi.it

## Abstract

This paper provides a comprehensive and computationally efficient case study for uncertainty quantification (UQ) and global sensitivity analysis (GSA) in a neuron model incorporating ion concentration dynamics. We address how challenges with UQ and GSA in this context can be approached and solved, including challenges related to computational cost, parameters affecting the system's resting state, and the presence of both fast and slow dynamics. Specifically, we analyze the electrodiffusive neuron-extracellular-glia (edNEG) model, which captures electrical potentials, ion concentrations ($Na^+$, $K^+$, $Ca^{2+}$, and $Cl^-$), and volume changes across six compartments. Our methodology includes a UQ procedure assessing the model's reliability and susceptibility to input uncertainty and a variance-based GSA identifying the most influential input parameters. To mitigate computational costs, we employ surrogate modeling techniques, optimized using efficient numerical integration methods. We propose a strategy for isolating parameters affecting the resting state and analyze the edNEG model dynamics under both physiological and pathological conditions. The influence of uncertain parameters on model outputs, particularly during spiking dynamics, is systematically explored. Rapid dynamics of membrane potentials necessitate a focus on informative spiking features, while slower variations in ion concentrations allow a meaningful study at each time point. Our study offers valuable guidelines for future UQ and GSA investigations on neuron models with ion concentration dynamics, contributing to the broader application of such models in computational neuroscience.

## Introduction

In the field of computational neuroscience, a multitude of models exists to describe neuronal activity, with ever-increasing biophysical intricacies [1]. Most of these models operate under the assumption of constant intra- and extracellular ion concentrations. This assumption is generally justified by the brain's inherent mechanisms, such as pumps and cotransporters, that work to uphold ion concentration levels close to baseline values. However, to investigate

**Data Availability Statement:** All relevant data are within the manuscript and its Supporting information files. The source codes for this study are available at https://github.com/CINPLA/

edNEGmodel and https://github.com/letiziasignorelli/edNEGmodel_UQSA.

**Funding:** MJS acknowledges support from the Research Council of Norway (Norges Forskningsråd: https://www.forskningsradet.no) via FRIPRO grant no 324239 (EMIx). The funders had no role in study design, data collection and analysis, decision to publish, or preparation of the manuscript.

**Competing interests:** The authors have declared that no competing interests exist.

scenarios where ion concentrations may change significantly—such as in epilepsy and spreading depression [2, 3]—certain models incorporate ion concentration dynamics and consider its influence on neuronal activity (see e.g., [4–21]).

This paper focuses on efficiently performing uncertainty quantification (UQ) and global sensitivity analysis (GSA) on neuron models incorporating ion concentration dynamics. Many model parameters inherently carry uncertainty arising from experimental measurement errors or physiological variations. For instance, the conductance $g_x$ of a specific ion channel $x$ is often uncertain. Despite the prevalence of such uncertainty, deterministic models assigning single numerical values to each parameter are common, and UQ and GSA have, to our knowledge, been overlooked or approached with simplistic one-at-a-time methods [19]. Understanding how uncertain parameters affect model outputs is crucial for extracting meaningful insights from the models [22–24]. Moreover, UQ and GSA serve as valuable tools to investigate channelopathies by revealing how modulations in specific ion channels or transporters influence neural activity. By conducting UQ and GSA on models with ion concentration dynamics, rather than more traditional models focused solely on membrane potentials, we can advance our understanding of such modulations. The conventional approach typically isolates the effects of channelopathies on neuronal membrane properties, overlooking their potential impact on the interstitial ion environment [25]. Neuron models with ion concentration dynamics enable us to examine the impact of channelopathies on ion concentrations, volume regulation, and neural activity patterns concurrently, providing a more comprehensive perspective. Furthermore, performing UQ and GSA under diverse conditions, such as normal spiking representing physiological states and activity indicating pathological states, would not only deepen our understanding of channelopathies, but also provide valuable insights into the underlying mechanisms associated with these varying conditions.

Performing UQ and GSA on neuron models with ion concentration dynamics is not straightforward. The focus of this paper is to provide general guidelines on how UQ and GSA can be performed on these types of models, and address the following main challenges: 1) the high computational cost associated with these models, 2) the influence of certain parameters on the resting state, and 3) the models' exhibition of both fast and slow dynamics. We will elaborate on these challenges in the following paragraphs.

The first challenge in performing UQ and GSA on models with ion concentration dynamics lies in the substantial computational cost involved. Neuron models with ion concentration dynamics are described by time-dependent, nonlinear, and strongly coupled ordinary differential equations (ODEs), or sometimes partial differential equations (PDEs). Their multi-scale nature introduces stiffness, leading to significant variations in time scales [26]. Simulating such models poses several numerical challenges arising from their inherent complexity. Capturing their dynamics precisely at different time scales is crucial for accurate simulations, but striking the right balance between accuracy and computational efficiency is challenging [27]. Moreover, calculating sensitivity indices within high-dimensional parameter spaces requires a considerable number of input-output evaluations. This challenge becomes particularly pronounced when dealing with models encompassing numerous state variables spanning several orders of magnitude and an expansive parameter space, as is often the case with neuron models incorporating ion concentration dynamics. To mitigate this challenge, the use of appropriate surrogate modeling techniques, such as polynomial chaos expansions (PCE) or reduced order models, becomes imperative for computational efficiency [22, 28–33]. These surrogate models enable the replacement of the original, computationally intensive model with a surrogate constructed from a relatively small experimental design. Subsequently, the surrogate can be utilized to compute sensitivity indices with minimal computational cost.

Another main challenge related to UQ and GSA on neuron models with ion concentration dynamics is that altering a membrane mechanism parameter may change the resting state of the system [19]. When the resting membrane potential is altered, it affects the driving force of all ion channels, subsequently impacting all active and passive currents and, consequently, the overall system dynamics. This inherent characteristic makes it difficult to distinguish the effect a specific parameter has on its associated membrane mechanism from the broader influence it exerts on all ion channels by altering the resting state.

The final challenge addressed in this paper pertains to the different temporal dynamics exhibited by the models' outputs, spanning both fast and slow time scales. These differences require careful consideration when determining which outputs to examine and in what manner. For instance, membrane potentials often display rapid dynamics characterized by multiple action potentials (AP). In such cases, it proves more meaningful to focus on informative spiking features, such as the number of APs, rather than the time-dependent membrane potential. Conversely, ion concentrations exhibit variations over slower time scales, allowing for a meaningful study at each point in time. Temporal factors influence the evolution of uncertainty over time, requiring an in-depth examination of how time interacts with uncertainty and sensitivity in time-dependent outputs [34]. Moreover, this challenge is linked to the first one concerning computational costs, as slow dynamics processes necessitate running longer simulations to observe their long-term effects. Additionally, evaluating sensitivity indices for a time-dependent output (i.e., one for each time point) is computationally demanding, especially when aiming at ensuring sufficient accuracy.

In this paper, we present a comprehensive and computationally efficient case study for UQ and GSA on a neuron model with ion concentration dynamics. Specifically, we delve into the electrodiffusive neuron-extracellular-glia (edNEG) model presented in Sætra et al. 2021 [20]. Electrodiffusive neuron models represent a subgroup of models with ion concentration dynamics that carefully account for ion concentrations by factoring in both diffusion and electric drift effects on ionic movement [19, 20]. In doing so, they maintain a consistent relationship between ion concentrations and electrical potentials, fostering a comprehensive understanding of the intricate interplay between ion dynamics and electric potentials. Our primary aim is to precisely assess the model's susceptibility to uncertainty, particularly during spiking dynamics, and compare it across physiological and pathological conditions. Thus, we specifically focus on the parameters of active ion channels, as they play a key role in the spiking activity of neurons. First, by considering effective implementation strategies and solver selection, we were able to run simulations that were 15 times more time-efficient than our original implementation presented in Sætra et al. 2021 [20]. This increased efficiency demonstrates the feasibility of conducting sensitivity analysis on complex neuroscience models. Second, to address the challenge associated with parameters potentially influencing the resting state of the system, we propose a method to isolate those parameters specifically affecting the resting state. This approach distinguishes parameters directly impacting system dynamics through active ion channels from those primarily affecting the resting state, subsequently shaping the system's overall dynamics. Finally, we investigate how model parameters influence ion dynamics and membrane potentials, considering both scalar quantities of interest and time-dependent outputs. This is achieved through a variance-based GSA, accounting for multiple parameters' simultaneous variation and interactions. We conduct this analysis on the edNEG model under both physiological and pathological conditions: first, when the neuron is subjected to a moderate stimulus current resulting in low-frequency firing, and second, when given a strong stimulus current inducing depolarization block. We believe our investigations can provide valuable guidelines for future studies on UQ and GSA applied to neuron models with ion concentration dynamics, thereby expanding the influence and application of such models in computational neuroscience.

The present study is organized as follows: First, we give a general introduction to the edNEG model, followed by a description of the numerical integration methods employed and the techniques applied for uncertainty quantification and sensitivity analysis. Next, we present the results from UQ and GSA. Finally, we provide discussions and future perspectives.

## Materials and methods

### The edNEG model: Mathematical and computational framework

The edNEG model [20] considers three domains—a neuron, extracellular space, and glia—each divided into two subdomains, resulting in a total of six compartments (Fig 1). Specifically, within the neuronal domain, the two compartments represent the somatic and dendritic layers, while the glial domain corresponds to a segment of astrocyte syncytium. Utilizing the electrodiffusive Kirchhoff-Nernst-Planck (KNP) framework, the model predicts the evolution in time of ion concentrations for four ion species ($Na^+$, $K^+$, $Ca^{2+}$, and $Cl^-$), the electrical potentials $\phi$, and the volumes $V$ in all compartments.

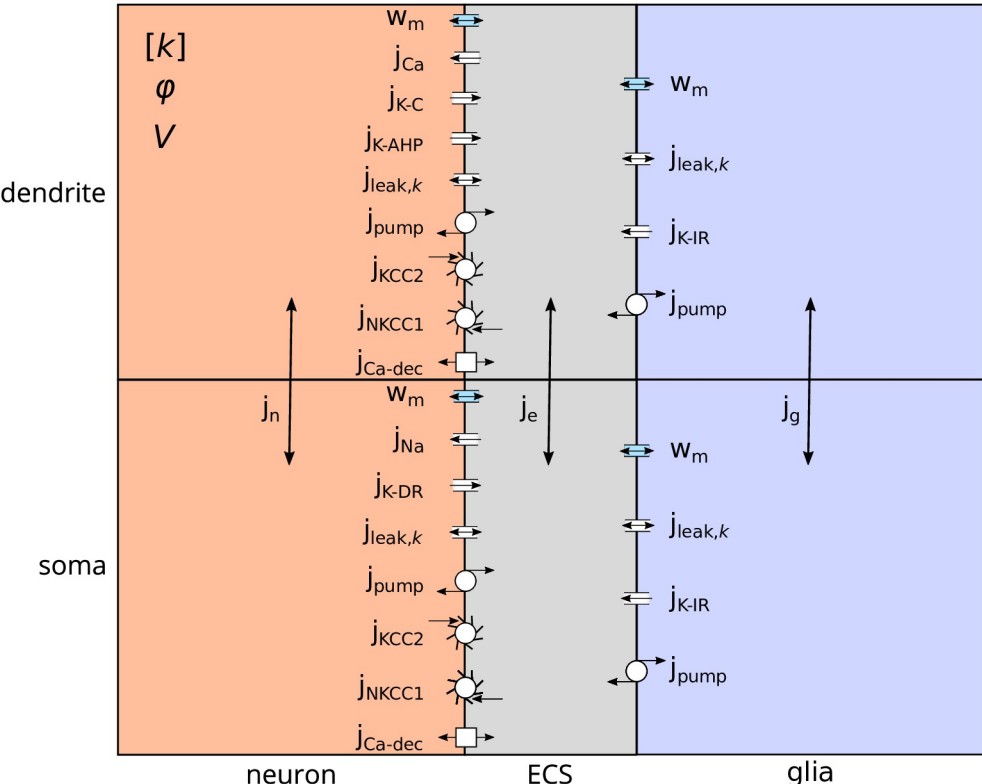

**Fig 1. edNEG model schematics.** The edNEG model comprises three domains representing a neuron, ECS, and glia, each subdivided into two compartments representing the somatic (bottom) and dendritic (top) layers. The model is embedded in the KNP framework and predicts the temporal evolution of the ion concentrations $[k]$ ($k = \{Na^+, K^+, Ca^{2+}, Cl^-\}$), the electrical potential $\phi$, and the volume $V$ in each compartment. Both neuronal membranes feature $Na^+$, $K^+$, and $Cl^-$ leak channels ($j_{leak,k}$), a $3Na^+/2K^+$ pump ($j_{pump}$), $K^+/Cl^-$ ($j_{KCC2}$) and $Na^+/K^+/2Cl^-$ ($j_{NKCC1}$) cotransporters, and a $Ca^{2+}/2Na^+$ exchanger ($j_{Ca-dec}$). Additionally, the soma contains $Na^+$ ($j_{Na}$) and $K^+$ delayed rectifier ($j_{K-DR}$) channels, while the dendrite includes a voltage-dependent $Ca^{2+}$ channel ($j_{Ca}$), a voltage-dependent $K^+$ afterhyperpolarization channel ($j_{K-AHP}$), and a $Ca^{2+}$-dependent $K^+$ channel ($j_{K-C}$). Both glial compartments feature $Na^+$ and $Cl^-$ leak channels ($j_{leak,k}$), an inward rectifying $K^+$ channel ($j_{K-IR}$), and a $3Na^+/2K^+$ pump ($j_{pump}$). Intra- and extracellular fluxes ($j_n$, $j_e$ and $j_g$) are driven by diffusion and electric drift. The model also accounts for transmembrane water flow ($w_m$) due to changes in the osmotic pressure across the membranes. The figure is adapted from Sætra et al. 2021 [20].

Interactions between the three domains are modeled by ionic exchange across the neuron-extracellular and glia-extracellular membranes, taking into account the distinct ion channels present in neuronal soma and dendrites, and in glial membrane. Specifically, in order to describe the transmembrane ion movements across the neuron-extracellular membrane, the following channels are considered: $Na^+$, $K^+$ and $Cl^-$ leak channels, a $3Na^+/2K^+$ pump, $K^+/Cl^-$ and $Na^+/K^+/2Cl^-$ cotransporters, and a $Ca^{2+}/2Na^+$ exchanger. Additionally, the soma compartment contains $Na^+$ and $K^+$ delayed rectifier channels, while the dendrite encloses a voltage-dependent $Ca^{2+}$ channel, a $K^+$ afterhyperpolarization channel, and a $Ca^{2+}$-dependent $K^+$ channel. Conversely, the glia-extracellular membrane includes $Na^+$ and $Cl^-$ leak channels, an inward rectifying $K^+$ channel, and a $3Na^+/2K^+$ pump. Movement of ions may also occur within domains, driven by diffusion and electric drift. The active ion channels are modeled through a Hodgkin-Huxley formalism for the voltage-dependent conductances, with six differential equations for the gating variables. Finally, volume dynamics induced by osmotic changes are described by six differential equations, each corresponding to a specific compartment. The ultimate model comprises $d = 34$ ODEs, with $d_N = 22$ addressing ion dynamics (six for each ion species, except for $Ca^{2+}$, which requires only four as it does not enter the glial compartments), $d_x = 6$ concerning gating variables, and $d_V = 6$ related to volume dynamics. This model can be represented by an ODE system in the form:

$$\begin{cases} \dot{\boldsymbol{N}}(t) & = f_N(\boldsymbol{N}(t), \boldsymbol{x}(t), \boldsymbol{V}(t), \boldsymbol{u}(t), \boldsymbol{p}), \qquad t \in (0, T], \\ \dot{\boldsymbol{x}}(t) & = f_x(\boldsymbol{N}(t), \boldsymbol{x}(t), \boldsymbol{V}(t), \boldsymbol{p}), \qquad t \in (0, T], \\ \dot{\boldsymbol{V}}(t) & = f_V(\boldsymbol{N}(t), \boldsymbol{V}(t), \boldsymbol{p}), \qquad t \in (0, T], \\ \boldsymbol{N}(0) & = \boldsymbol{N}_0, \\ \boldsymbol{x}(0) & = \boldsymbol{x}_0, \\ \boldsymbol{V}(0) & = \boldsymbol{V}_0. \end{cases} \qquad (1)$$

Here, $\boldsymbol{N}(t) = (N_1(t), ..., N_{d_N}(t))' \in \mathbb{R}^{d_N}$ is the ion dynamics state vector, where $\boldsymbol{N}$ are the number of ions, $\boldsymbol{x}(t) = (x_1(t), ..., x_{d_x}(t))' \in \mathbb{R}^{d_x}$ is the gating variables state vector, and $\boldsymbol{V}(t) = (V_1(t), ..., V_{d_V}(t))' \in \mathbb{R}^{d_V}$ is the volume dynamics state vector. $\boldsymbol{u}(t) \in \mathbb{R}^{d_N}$ is a known system input vector, specifically a stimulus current in our case, $\boldsymbol{p} \in \mathbb{R}^{N_p}$ is a vector of known parameters, $f_N : \mathbb{R}^{d \times N_p} \mapsto \mathbb{R}^{d_N}$, $f_x : \mathbb{R}^{d \times N_p} \mapsto \mathbb{R}^{d_x}$, $f_V : \mathbb{R}^{d \times N_p} \mapsto \mathbb{R}^{d_V}$ is a set of coupled nonlinear functions, and $\boldsymbol{N}_0 \in \mathbb{R}^{d_N}$, $\boldsymbol{x}_0 \in \mathbb{R}^{d_x}$, $\boldsymbol{V}_0 \in \mathbb{R}^{d_V}$ represent the initial conditions of the system. The six electrical potentials are derived algebraically at each time step, while the four membrane potentials are defined as the difference between intra- and extracellular electrical potentials, both for neuronal and glial cells in each of their two compartments. Subsequently, we can establish the set of output equations as follows:

$$\boldsymbol{\phi}(t) = g(\boldsymbol{N}(t), \boldsymbol{V}(t), \boldsymbol{p}), \qquad t \in (0, T], \qquad (2)$$

where $\boldsymbol{\phi}(t) \in \mathbb{R}^{d_\phi}$, $d_\phi = 10$ are the outputs. The detailed mathematical formulation is reported in Sætra et al. 2021 [20].

## Numerical implementation and validation

Approximating, simulating, and validating multi-scale biophysical models such as the edNEG model poses several numerical challenges. Furthermore, it is crucial to minimize the

computational cost needed for model execution to facilitate efficient UQ and GSA within a reasonable timeframe. In reference to the implementation proposed in Sætra et al. 2021 [20], where we utilized the `solve_ivp` function from the Python library `SciPy` [35] and its RK23 method, we successfully improved convergence by rescaling units and implementing an analytical Jacobian. The first strategy aimed at confining the range of orders of magnitude for the state variables. The second strategy sped up computations and improved result accuracy. Specifically, the analytical Jacobian prevented approximation errors resulting from SciPy's default finite difference approximation. These changes made implicit solvers work effectively, and resulted in simulations that were up to fifteen times faster than the original ones, depending on the choice of integration method and maximum time step. We subsequently conducted a convergence analysis of the number of action potentials and time of the last action potential for $\phi_{msn}$, the extracellular potassium concentration $[K^+]_{se}$, and the extracellular volume $V_{se}$ for `solve_ivp`'s different solvers. As a result of this analysis, for UQ and GSA, we opted for the implicit solver Radau with a maximum time-step length of $\Delta t_{max} = 10$ ms. Note that the `solve_ivp` function utilizes adaptive time stepping, and $\Delta t_{max}$ indicates the maximum allowed time step. Our choice balanced computational efficiency with the necessity to effectively capture dynamics within a simulation time of T = 6 s. Our selection of the Radau solver was primarily due to its implicit nature, which allows accurate results to be obtained with larger time-step lengths. Additionally, it proved to be more accurate at large time-step lengths for our particular model compared to other `SciPy` implicit solvers.

The complete set of rescaled parameters along with initial conditions are summarized in S1 Appendix. The source codes for this study are available at https://github.com/CINPLA/edNEGmodel and https://github.com/letiziasignorelli/edNEGmodel_UQSA [36].

## Variance-based global sensitivity analysis

Let us call a computational model $\mathcal{G}$, i.e. a six-compartmental neuron model, that depends on time $t$ and has $N_p$ uncertain input parameters $\mathbf{p} = (p_1, p_2, \ldots, p_{N_p})$, with output $y$:

$$\mathbf{p} \in \mathcal{D}_\mathbf{p} \in \mathbb{R}^{N_p} \mapsto y = \mathcal{G}(t, \mathbf{p}) \in \mathbb{R}, \tag{3}$$

where $\mathcal{D}_\mathbf{p}$ denotes the support of the set of $N_p$ input parameters. To address the uncertainties associated with the input, the computational model can be analyzed using a probabilistic approach. Therefore, suppose that the uncertainty in the input parameters is modeled by a random vector $\mathbf{P} = [P_1, P_2, \ldots, P_{N_p}]$ with prescribed joint probability density function $f_\mathbf{P}(\mathbf{p})$. The resulting quantity of interest $Y$ is now a random variable obtained by propagating the uncertainty in $\mathbf{P}$ through $\mathcal{G}$:

$$Y = \mathcal{G}(t, \mathbf{P}) \in \mathbb{R}. \tag{4}$$

Variance-based GSA provides valuable information about which parameters have the most significant influence on the output variability and uncovers potential interactions among these parameters. One widely used approach within variance-based GSA is the computation of Sobol' sensitivity indices [37, 38]. Sobol' indices are a set of sensitivity indices that partition the total variance of the model output into contributions from individual parameters and their interactions. Considering the quantity of interest $Y = \mathcal{G}(t, \mathbf{P})$ as defined in Eq 4, let us fix the factor $P_i$ at a particular value $p_i^*$. The resulting variance of the model output, with $P_i$ fixed, will measure the relative importance of $P_i$ and can be defined as follows:

$$\mathrm{Var}_{\mathbf{P}_{\sim i}}[Y | P_i = p_i^*]. \tag{5}$$

Taking the average and according to the *law of total variance*, the variance of the whole model output can be decomposed as

$$\text{Var}[Y] = \mathbb{E}_{P_i}[\text{Var}_{\mathbf{P}_{\sim i}}[Y|P_i]] + \text{Var}_{P_i}[\mathbb{E}_{\mathbf{P}_{\sim i}}[Y|P_i]]. \tag{6}$$

The conditional variance $\text{Var}_{P_i}[\mathbb{E}_{\mathbf{P}_{\sim i}}[Y|P_i]]$ is defined as the *first-order effect* of $P_i$ on $Y$. Finally, the First Order Sobol' sensitivity index of the input $P_i$ on $Y$ is given by

$$S_i = \frac{\text{Var}_{P_i}[\mathbb{E}_{\mathbf{P}_{\sim i}}[Y|P_i]]}{\text{Var}[Y]}, \qquad i = 1, \ldots, N_p, \tag{7}$$

and is interpreted as the fraction of the output variance that can be associated to the variance of $P_i$. A high value of $S_i$ indicates that changes in parameter $P_i$ have a significant impact on the output, while a low value suggests that variations in $P_i$ have a relatively minor effect on the overall output variance.

First Order Sobol' indices do not consider potential interactions among parameters, a common occurrence in neuroscience models. To address this limitation, a possible approach is to compute the *total effects*. We define the variance of the expected value when all parameters except $P_i$ are fixed as

$$\text{Var}_{\mathbf{P}_{\sim i}}[\mathbb{E}_{P_i}[Y|\mathbf{P}_{\sim i}]]. \tag{8}$$

Therefore, due to the *law of total variance*, the Total Order Sobol' sensitivity of $P_i$ on $Y$ is given by

$$S_{T_i} = 1 - \frac{\text{Var}_{\mathbf{P}_{\sim i}}[\mathbb{E}_{P_i}[Y|\mathbf{P}_{\sim i}]]}{\text{Var}[Y]} = \frac{\mathbb{E}_{\mathbf{P}_{\sim i}}[\text{Var}_{P_i}[Y|\mathbf{P}_{\sim i}]]}{\text{Var}[Y]}, \qquad i = 1, \ldots, N_p. \tag{9}$$

$S_{T_i}$ provides a measure of the output variance attributed to $P_i$ considering all possible interactions of any order with any other parameter.

**The case of time-dependent processes.** Many neuroscience models, such as the edNEG model examined in this study, exhibit time-dependent outputs, yielding additional challenges for sensitivity analysis. While it is possible to apply Sobol's approach pointwise in time, for instance on a set of grid points, $0 = t_0 < t_1 < \ldots < t_{n-1} < t_n = T$, this approach has some limitations. First, the variance of the process itself varies in time, leading to distortions in the evaluation of the relative importance of uncertain parameters over different time periods. Second, considering the model outputs at different time points as independent variables ignores the temporal correlation structure of the process. To overcome the first issue, one possible option is to compute a weighted Sobol' indices pointwise in time, where at each time $t_k$ the Sobol' index is multiplied by the standard deviation of the output $Y = \mathcal{G}(t, \mathbf{P})$:

$$S_{T_i}^W(\mathcal{G}; t) = S_{T_i}(\mathcal{G}; t)\sqrt{\text{Var}[\mathcal{G}(t)]}. \tag{10}$$

A further improvement could involve incorporating a generalized version of Sobol' indices considering potential time correlations [34]. The generalized First Order Sobol' indices read as

$$\mathfrak{S}_i(\mathcal{G}; T) = \frac{\int_0^T \text{Var}_{P_i}[\mathbb{E}_{\mathbf{P}_{\sim i}}[\mathcal{G}|P_i](t)]dt}{\int_0^T \text{Var}[\mathcal{G}(t)]dt}, \qquad i = 1, \ldots, N_p, \tag{11}$$

**Table 1. "Non-dynamic" parameters: Conductances and strengths of leak channels and homeostatic mechanisms.**

| Parameter | | Nominal value | Units |
|---|---|---|---|
| $\bar{g}_{\text{Na,leak,n}}$ | Na$^+$ neuron leak conductance | 0.0246 | mS $\cdot$ cm$^{-2}$ |
| $\bar{g}_{\text{K,leak,n}}$ | K$^+$ neuron leak conductance | 0.0245 | mS $\cdot$ cm$^{-2}$ |
| $\bar{g}_{\text{Cl,leak,n}}$ | Cl$^-$ neuron leak conductance | 0.1 | mS $\cdot$ cm$^{-2}$ |
| $\rho_{\text{n}}$ | Na$^+$/K$^+$ neuron pump strength | $1.87 \times 10^{-4}$ | nmol $\cdot$ cm$^{-2}$ $\cdot$ ms$^{-1}$ |
| $U_{\text{kcc2}}$ | KCC2 cotransporter strength | $1.49 \times 10^{-5}$ | nmol $\cdot$ cm$^{-2}$ $\cdot$ ms$^{-1}$ |
| $U_{\text{nkcc1}}$ | NKCC1 cotransporter strength | $2.33 \times 10^{-5}$ | nmol $\cdot$ cm$^{-2}$ $\cdot$ ms$^{-1}$ |
| $U_{\text{Ca-dec}}$ | Ca$^{2+}$ decay rate | 0.075 | ms$^{-1}$ |
| $\bar{g}_{\text{Na,leak,g}}$ | Na$^+$ glial leak conductance | 0.1 | mS $\cdot$ cm$^{-2}$ |
| $\bar{g}_{\text{K-IR}}$ | K$^+$ glial leak conductance | 1.696 | mS $\cdot$ cm$^{-2}$ |
| $\bar{g}_{\text{Cl,leak,g}}$ | Cl$^-$ glial leak conductance | 0.01 | mS $\cdot$ cm$^{-2}$ |
| $\rho_{\text{g}}$ | Na$^+$/K$^+$ glial pump strength | $1.12 \times 10^{-4}$ | nmol $\cdot$ cm$^{-2}$ $\cdot$ ms$^{-1}$ |

while the generalized Total Order Sobol' indices are:

$$\mathfrak{S}_{T_i}(\mathcal{G}; T) = 1 - \frac{\int_0^T \text{Var}_{\mathbf{P}_{\sim i}}[\mathbb{E}_{P_i}[\mathcal{G}|\mathbf{P}_{\sim i}](t)]dt}{\int_0^T \text{Var}[\mathcal{G}(t)]dt}, \qquad i = 1, \ldots, N_p. \tag{12}$$

## Factor fixing

**Procedure description.** Based on the edNEG model's biophysics, and informed by a preliminary and previously done sensitivity analysis of the analogous edPR model [39], we made an educated guess to categorize the parameters of interest into two distinct groups. The first group influences the model's dynamics by altering the resting state, while the second group directly impacts the dynamics themselves. We have labeled these groups as "non-dynamic" parameters and "dynamic" parameters, respectively. The nominal values and the detailed description for these parameters are listed in Tables 1 and 2. To ensure that only the "non-dynamic" parameters influence the resting state and subsequently modify the dynamics, we carried out a sensitivity analysis by treating the two parameter groups as two inputs of uncertainty. Consequently, we computed the Total Order Sobol' indices with respect to both groups of parameters. The aim of this factor fixing procedure was to ensure that the "dynamic" parameters did not affect the resting state, allowing us to subsequently concentrate on examining how uncertainty in these parameters uniquely influences firing dynamics.

**Parameter distributions.** We adopted the assumption that all parameters under consideration for the sensitivity analysis followed a uniform distribution [23] within a predefined percentage range centered around their nominal values. For this reason a 'hyper-parameter' $\hat{\sigma}$ was introduced to control the uncertainty across all parameters, as in Pathmanathan et al.

**Table 2. "Dynamic" parameters: Maximum conductances of active channels.**

| Parameter | | Nominal value | Units |
|---|---|---|---|
| $\bar{g}_{\text{Na}}$ | maximum conductance of Na$^+$ current | 30 | mS $\cdot$ cm$^{-2}$ |
| $\bar{g}_{\text{DR}}$ | maximum conductance of K$^+$ delayed rectifier current | 15 | mS $\cdot$ cm$^{-2}$ |
| $\bar{g}_{\text{Ca}}$ | maximum conductance of Ca$^{2+}$ current | 11.8 | mS $\cdot$ cm$^{-2}$ |
| $\bar{g}_{\text{AHP}}$ | maximum conductance of AHP current | 0.8 | mS $\cdot$ cm$^{-2}$ |
| $\bar{g}_{\text{C}}$ | maximum conductance of Ca$^{2+}$-dependent K$^+$ current | 15 | mS $\cdot$ cm$^{-2}$ |

2019 [24]:

$$P_i \sim \mathcal{U}(P_{i,\text{nom}} - |\hat{\sigma} P_{i,\text{nom}}|, P_{i,\text{nom}} + |\hat{\sigma} P_{i,\text{nom}}|), \qquad \text{for } i \in \{1, 2, \ldots, N_p\}, \qquad (13)$$

where $P_{i,\text{nom}}$ is the nominal value of $P_i$. In the factor fixing analysis, $\hat{\sigma}$ was established at a value of 15%.

**Simulation protocol.** To evaluate the parameters' impact on the resting state, we conducted the GSA on simulations that we ran for 240 s without applying any stimulus.

**Quantities of interest.** We opted to analyze six representative variables:

- $\phi_{\text{msn}}$: neuronal membrane potential (soma layer),

- $\phi_{\text{mdn}}$: neuronal membrane potential (dendrite layer),

- $\phi_{\text{msg}}$: glial membrane potential (soma layer),

- $\phi_{\text{mdg}}$: glial membrane potential (dendrite layer),

- $[\text{K}^+]_{\text{se}}$: extracellular potassium concentration (soma layer),

- $[\text{K}^+]_{\text{de}}$: extracellular potassium concentration (dendrite layer),

and designated as the quantities of interest (QoI) their values at the simulation's final time. Subsequently, we computed the Total Order Sobol' indices for each of them. We selected these QoIs because temporally constant membrane potentials and ion concentrations characterize the system's resting state. If a parameter is altered, the membrane potentials and ion concentrations may deviate from an initial resting state until the system stabilizes in a new resting state.

**Implementation details.** We employed the Python library `SaLib` [40, 41] to create a set of parameter configurations using the Saltelli method. This approach extends the Sobol' sequence, a widely utilized quasi-random low-discrepancy sequence, to generate uniform samples across the parameter space. This extension aims to minimize error rates in the subsequent sensitivity index computations [42]. Our study employed 512 samples, which translated to 3072 parameter sets. Since the factor-fixing analysis was a preliminary analysis conducted with the model in resting conditions and with scalar outputs, it did not require the use of surrogate models.

## Uncertainty quantification and sensitivity analysis in the dynamical state

**Procedure description.** Our second analysis focused on the "dynamic" parameters group, specifically concentrating on the five parameters listed in Table 2. This approach enabled us to conduct an in-depth exploration of the specific effects these "dynamic" parameters exert on the system's behavior, free from the influence of the "non-dynamic" parameters group altering the resting state.

**Parameter distributions.** As in the factor fixing analysis, we defined the distributions of input parameters as uniform distributions. To parameterize these distributions, we set the hyper-parameter $\hat{\sigma}$ to 5%. Subsequently, we changed the uncertainty level in order to study the model's dependence on input uncertainty, specifically we chose $\hat{\sigma} \in \{0, 1, 5, 10\}\%$.

**Simulation protocol.** The evaluation of UQ and GSA for the dynamical state was conducted on two representative simulations during firing activity. In the first case, the model was subject to a stimulus current defined as $I_{\text{stim}}(t) = 8 \times 10^{-5} \, 1_{(0.2, 5.5)\text{s}}(t) \, \mu\text{A}$, with T = 6 s, simulating physiological conditions. In contrast, the second scenario investigated pathological conditions, employing a stimulus current defined as $I_{\text{stim}}(t) = 20 \times 10^{-5} \, 1_{(0.2, 5.5)\text{s}}(t) \, \mu\text{A}$, with T = 6 s.

The neuron was stimulated by introducing a $K^+$ injection current into the somatic compartment. To maintain ion conservation, an equivalent amount of ions was removed from the corresponding somatic extracellular compartment [20].

**Quantities of interest.** Given the greater complexity of the dynamics and the increased number of parameters involved in this context, a more careful selection process is required. To address this challenge, we have opted to analyze only two representative state variables: the transmembrane potential $\phi_{msn}$ and the ion concentration $[K^+]_{se}$. Proper configurations are essential for each variable. Due to the rapid dynamics of $\phi_{msn}$, conducting an uncertainty quantification and sensitivity analysis at every time-step might not yield meaningful results. In scenarios like this, a common practice is to identify a few key scalar spiking features as relevant QoIs, as seen in previous works such as Pathmanathan et al. 2019 [24] and Ghori and Kang 2023 [23], and compute the Total Order Sobol' indices for each of them. Consequently, in the physiological case we have identified the following three main features to serve as our outputs:

- $N_{AP}$ : number of action potentials (AP),

- $f_{final}$ : final firing frequency in the last 1.5 s of firing activity (that is, for $t \in (4.0, 5.5)$ s),

- $T_{bFAP}$ : time before the first AP, that is, the duration between when a neuron is subjected to a stimulus and the moment it generates its initial spike.

    Conversely, the two key features in the pathological case are the following:

- $T_{sDP}$ : time of the start of the depolarization block, which was defined as the time when the first derivative of $\phi_{msn}$ becomes consistently constant and falls below a predefined threshold level. In our implementation, this threshold was set at $10^{-1}$,

- $T_{bFAP}$ : time before the first AP.

    Moreover, for the second chosen variable, $[K^+]_{se}$, the situation is quite different. Given its slower dynamics, it is indeed meaningful and informative to conduct uncertainty quantification and sensitivity analysis at each time-step. Consequently, in both test cases, we compute the Total Order Sobol' indices for every time step. Furthermore, we have implemented the enhancements outlined in the section titled The case of time-dependent processes, which include the computation of weighted Total Order Sobol' indices, $S_{T_i}^W(\mathcal{G}; t)$, and of generalized Total Order Sobol' indices, $\mathfrak{S}_{T_i}(\mathcal{G}; T)$.

    The choice of QoIs will naturally depend on the type of study being conducted. Since the goal of this study is to demonstrate how to perform UQ and GSA on neuron models with ion concentration dynamics, we choose to focus on general spiking features capturing the neuronal firing pattern when studying the membrane potential. To represent slow dynamics, we select $[K^+]_{se}$ because of its crucial role in depolarization blocks [19]. For those interested in exploring other spiking features such as the width or height of the APs, or slow dynamics variables like other ion concentrations or volumes, they can readily apply the same methodology that we have used.

**Implementation details.** We employed the Python library `Uncertainpy` [22], a toolbox designed for uncertainty quantification and sensitivity analysis specifically tailored for computational neuroscience. To enhance computational efficiency, we opted for a Polynomial Chaos Expansion (PCE) [28–30] of our model using polynomials of order 4, employing the Point Collocation method. For more information, see Tennøe et al. 2018 [22], the Uncertainpy documentation (https://uncertainpy.readthedocs.io/), and Feinberg and Langtangen 2015 [43]. The UQ and GSA runtimes varied between 30 minutes for physiological conditions with scalar output and 120 minutes for pathological conditions with time-dependent output.

Timings were conducted on an Acer SPIN 5 SP513–52N with an Intel Core i5–8250U CPU running at 1.60GHz and 4 cores, using Python 3.8.

## Results

### edNEG modeling of physiological and pathological activity

In the absence of any stimulus, the edNEG model maintains a stable resting state achieved through a delicate balance among ion-specific leakage channels, pumps, and cotransporters. During this resting phase, ion concentrations, membrane potential, and volume values remain constant, aligning with their initial conditions. At moderately low current injections, physiological activity emerges with corresponding modest firing rates (Fig 2A, 2C and 2E). Over extended simulation time, this leads the model into a dynamic steady state, supporting sustained firing over an extended period without significant ion concentration divergence from baseline values [20].

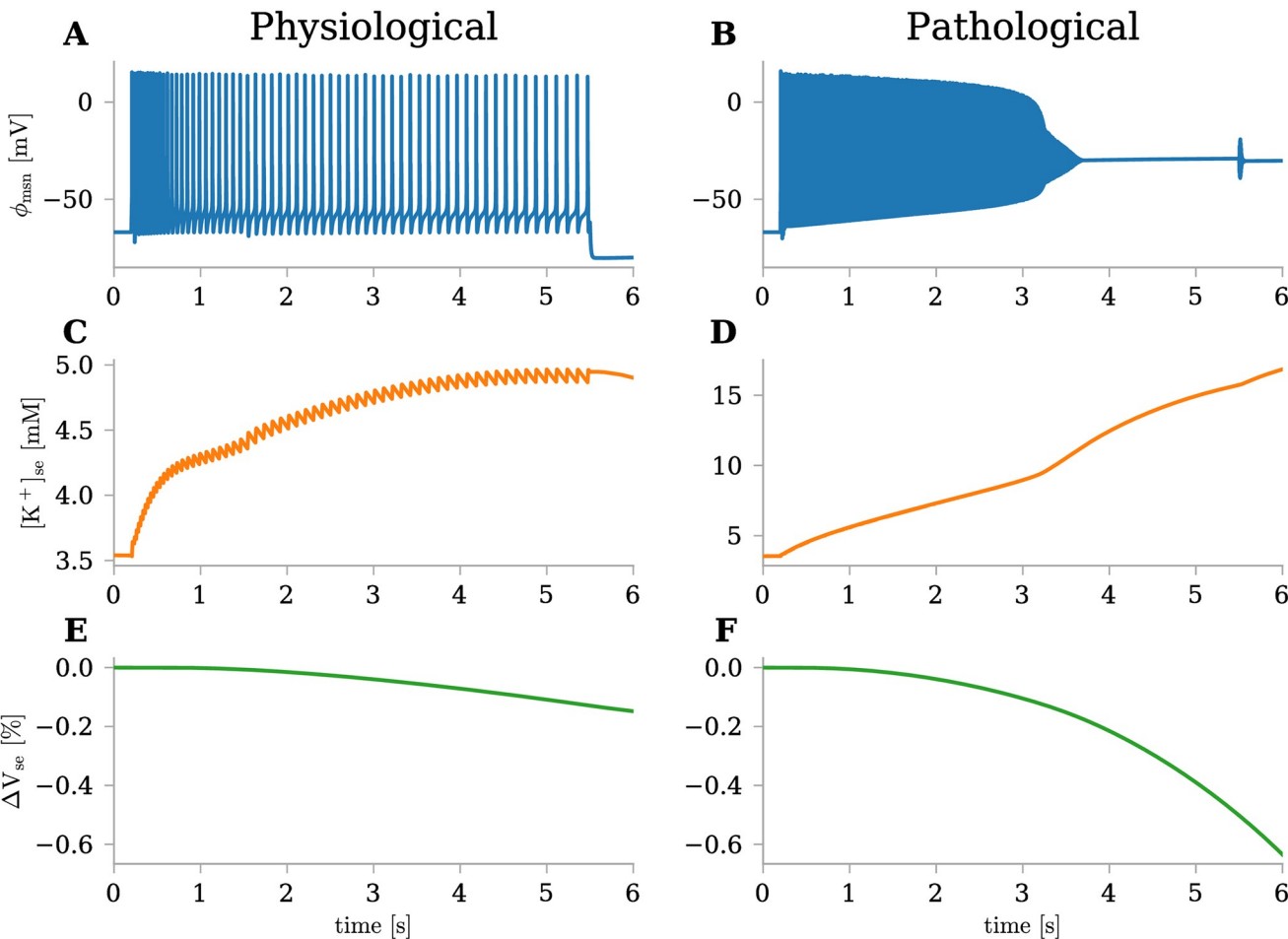

**Fig 2. edNEG modeling of physiological and pathological activity.** Model's response to different stimulus strenghts, illustrated by three representative state variables: membrane potential of neuronal soma ($\phi_{msn}$), potassium concentration in extracellular space outside soma ($[K^+]_{se}$), and deviance from baseline values of the extracellular space volume outside soma ($\Delta V_{se}$). **(A,C,E)** Physiological response to a $8 \times 10^{-5}$ $\mu$A stimulus current applied to the somatic compartment between $t = 0.2$ s and $t = 5.5$ s. **(B,D,F)** Pathological response to a $20 \times 10^{-5}\mu$A stimulus current applied to the somatic compartment between $t = 0.2$ s and $t = 5.5$ s. The edNEG model was originally presented in Sætra et al. 2021 [20].

**Table 3. Total order Sobol' indices for six QoIs with $\hat{\sigma} = 15\%$.**

| Output variables | $S_{T_i}^{\text{non-dynamic}}$ | $S_{T_i}^{\text{dynamic}}$ |
|---|---|---|
| $\phi_{\text{msn}}$ | 0.99 | $3.3 \times 10^{-3}$ |
| $\phi_{\text{mdn}}$ | 0.99 | $3.3 \times 10^{-3}$ |
| $\phi_{\text{msg}}$ | 0.99 | $6.7 \times 10^{-5}$ |
| $\phi_{\text{mdg}}$ | 0.99 | $7.1 \times 10^{-5}$ |
| $[\text{K}^+]_{\text{se}}$ | 0.99 | $2.7 \times 10^{-4}$ |
| $[\text{K}^+]_{\text{de}}$ | 0.99 | $4.4 \times 10^{-4}$ |

Pathological activity arises when stabilizing mechanisms fail to keep pace with ionic exchange through active ion channels (Fig 2B, 2D and 2F). This occurs under strong stimulus currents, resulting in elevated firing rates. During firing, ion concentrations gradually change, resulting in a progressive depolarization of the neuron, leading to even faster firing. However, the neuron's ability to tolerate this intense input is limited, and after a few seconds, it becomes unable to repolarize, causing firing to cease. In neuroscience, this condition, where a neuron is depolarized to a voltage level that renders it incapable of generating further action potentials, is referred to as "depolarization block" [20, 44].

## Factor fixing: "Non-dynamic" parameters dominate the resting state

To select the subset of parameters that have minimal impact on changes in the resting state, we conducted a sensitivity analysis employing Total Order Sobol' indices $S_{T_i}$ (Table 3) with respect to each of the two parameters groups listed in Tables 1 and 2. This analysis confirms that the "non-dynamic" parameters group significantly impacts the uncertainty of the chosen model outputs, exceeding the influence of the "dynamic" parameters group, as we expected. Specifically, we observed the value of $S_{T_i}$ being fixed at 0.99 for all output variables, indicating a pronounced effect of the "non-dynamic" parameters group on the resting state.

## Uncertainty quantification and sensitivity analysis under physiological conditions

**All selected QoIs are influenced by input uncertainty at comparable levels.** In this section, we present the findings from our analysis concerning the uncertainty of the state variable $\phi_{\text{msn}}$ under physiological conditions. This involves the investigation of selected QoIs, including the number of action potentials ($N_{\text{AP}}$), the final firing frequency ($f_{\text{final}}$), and the time before the first action potential ($T_{\text{bFAP}}$). We began by setting the uncertainty at a moderate level of $\hat{\sigma} = 5\%$. This allowed us to obtain a broad understanding of the degree of uncertainty present in the model outputs. Histograms in Fig 3A–3C are representative of approximate probability density functions for the three QoIs. The figures indicate that $N_{\text{AP}}$ spans approximately from 50 to 90 action potentials, $f_{\text{final}}$ ranges between 6.0 and 12.0 Hz, and $T_{\text{bFAP}}$ falls within the range of 7.8 to 8.2 ms. The variability in $N_{\text{AP}}$ and $f_{\text{final}}$ is meaningful; although many values cluster around the baseline, there is a substantial spread. This wide range signifies that uncertainties in input parameters have a pronounced impact on these particular QoIs. Regarding $T_{\text{bFAP}}$, its range is slightly narrower—about 0.4 ms. Notably, both $N_{\text{AP}}$ and $f_{\text{final}}$ exhibit a Gaussian-like distribution, with their baseline values aligning closely with the respective distribution modes. Conversely, $T_{\text{bFAP}}$ demonstrates a more uniform distribution.

To further investigate the model's robustness to parameter uncertainty, we changed the uncertainty levels in our simulations (Fig 3D–3G). In line with our earlier observations, $N_{\text{AP}}$

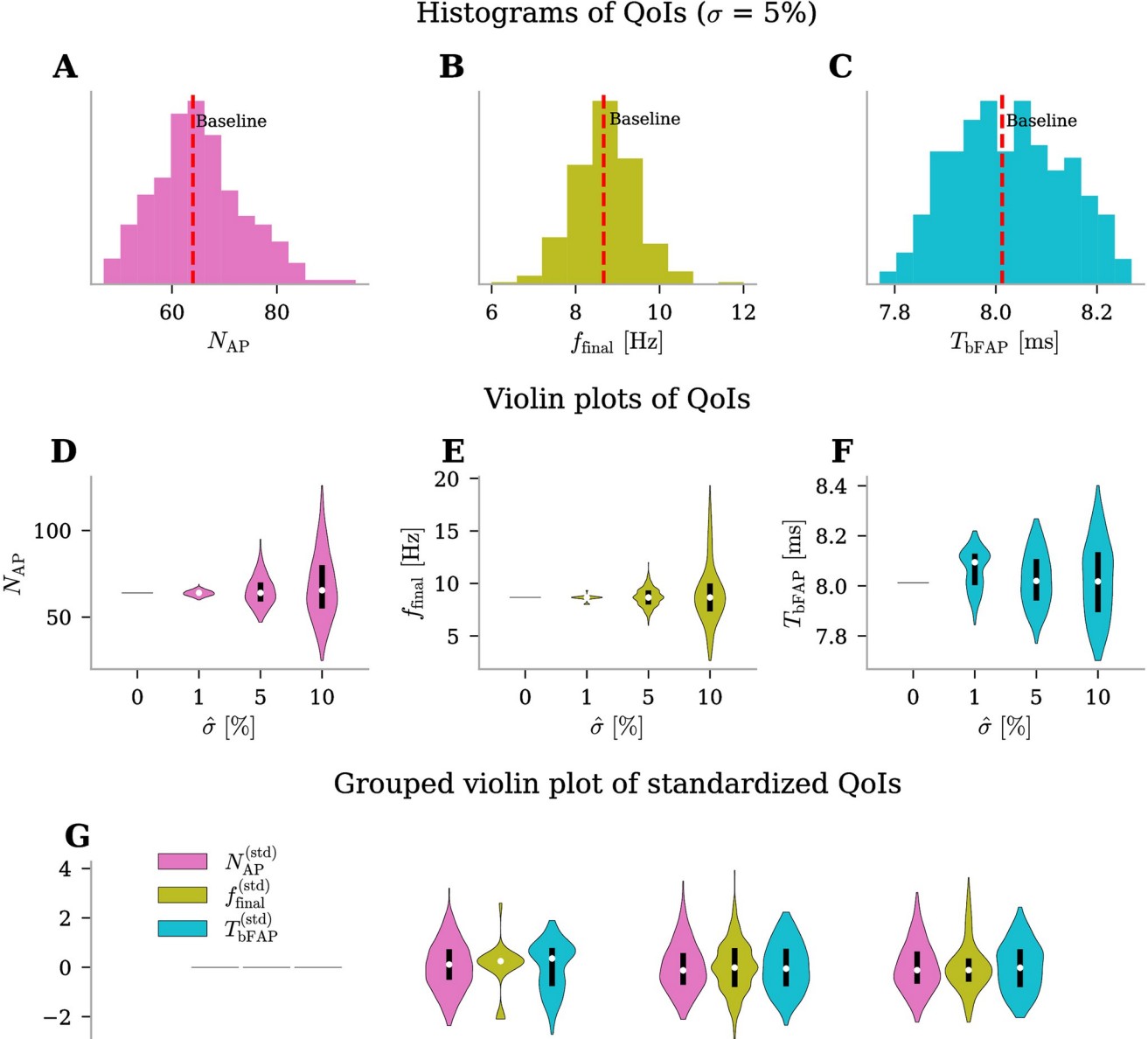

**Fig 3. Uncertainty quantification for $\phi_{\mathbf{msn}}$ QoIs under physiological conditions.** Outputs for three QoIs (number of action potentials, $N_{\mathrm{AP}}$, final firing frequency, $f_{\mathrm{final}}$, and time before the first action potential, $T_{\mathrm{bFAP}}$). **(A-C)** Histograms of QoIs with moderate uncertainty level on input parameters, $\hat{\sigma} = 5\%$. The red dashed lines indicate the values of QoIs when the edNEG model is evaluated using baseline parameter values. **(D-F)** Violin plots of QoIs with increasing level of uncertainty in the input parameters, $\hat{\sigma} = \{0, 1, 5, 10\}\%$ (x-axis). The width of each violin at any given point indicates the density of the QoI at the corresponding uncertainty level. The white dots represent the median and the black bold lines indicate the data that lie within quartiles. **(G)** Grouped violin plot of standardized QoIs ((data—mean)/standard deviation) with increasing level of uncertainty in the input parameters.

and $f_{\mathrm{final}}$ are considerably influenced by input uncertainty. As the level of uncertainty increases, the distributions of $N_{\mathrm{AP}}$ and $f_{\mathrm{final}}$ broaden. $N_{\mathrm{AP}}$ transitions from a confined range around 64 at $\hat{\sigma} = 1\%$ to spanning between 25 and 120 action potentials at $\hat{\sigma} = 10\%$ (Fig 3D). Furthermore, at $\hat{\sigma} = 1\%$, also $f_{\mathrm{final}}$ exhibits minimal sensitivity to input uncertainty, with

values closely clustered around the baseline of 8.67 Hz (Fig 3E). However, this dependence escalates as $\hat{\sigma}$ attains higher values. Moreover, $T_{bFAP}$ shows a moderate amplification in output uncertainty, evident by the broader distribution as uncertainty increases, even though less pronounced than the effect on $N_{AP}$ and $f_{final}$ (Fig 3F). Lastly, in Fig 3G, we depict standardized QoI distributions, categorized by distinct uncertainty levels. This allows us to compare the impact of uncertainty on the different QoIs. It is now evident that all QoIs are influenced by input uncertainty at comparable levels. However, the distributions of both $N_{AP}$ and $f_{final}$ exhibit a Gaussian-like shape that is slightly more widely dispersed and includes a few additional outliers. In contrast, $T_{bFAP}$ displays a more evenly distributed range of output values.

**The selected QoIs are influenced in distinct ways by the five uncertain parameters.** In order to understand which of the input parameters predominantly influence the uncertainty in the QoIs linked to $\phi_{msn}$, we carried out a sensitivity analysis employing First and Total Order Sobol' indices (Fig 4). One immediate observation from comparing the first and total effects is that the majority of the uncertainty in $N_{AP}$ and $f_{final}$ arises from first-order effects, evidenced by the negligible difference between their First and Total Order indices. Conversely, concerning $T_{bFAP}$, a substantial difference is evident, signifying that its uncertainty is largely attributed to interactions among input parameters. Delving into the individual analysis of the three QoIs, findings for $N_{AP}$ (Fig 4A and 4D) and $f_{final}$ (Fig 4B and 4E) are very similar, and imply that these QoIs are most sensitive to the input parameters $g_{DR}$, $g_C$, and $g_C$. These

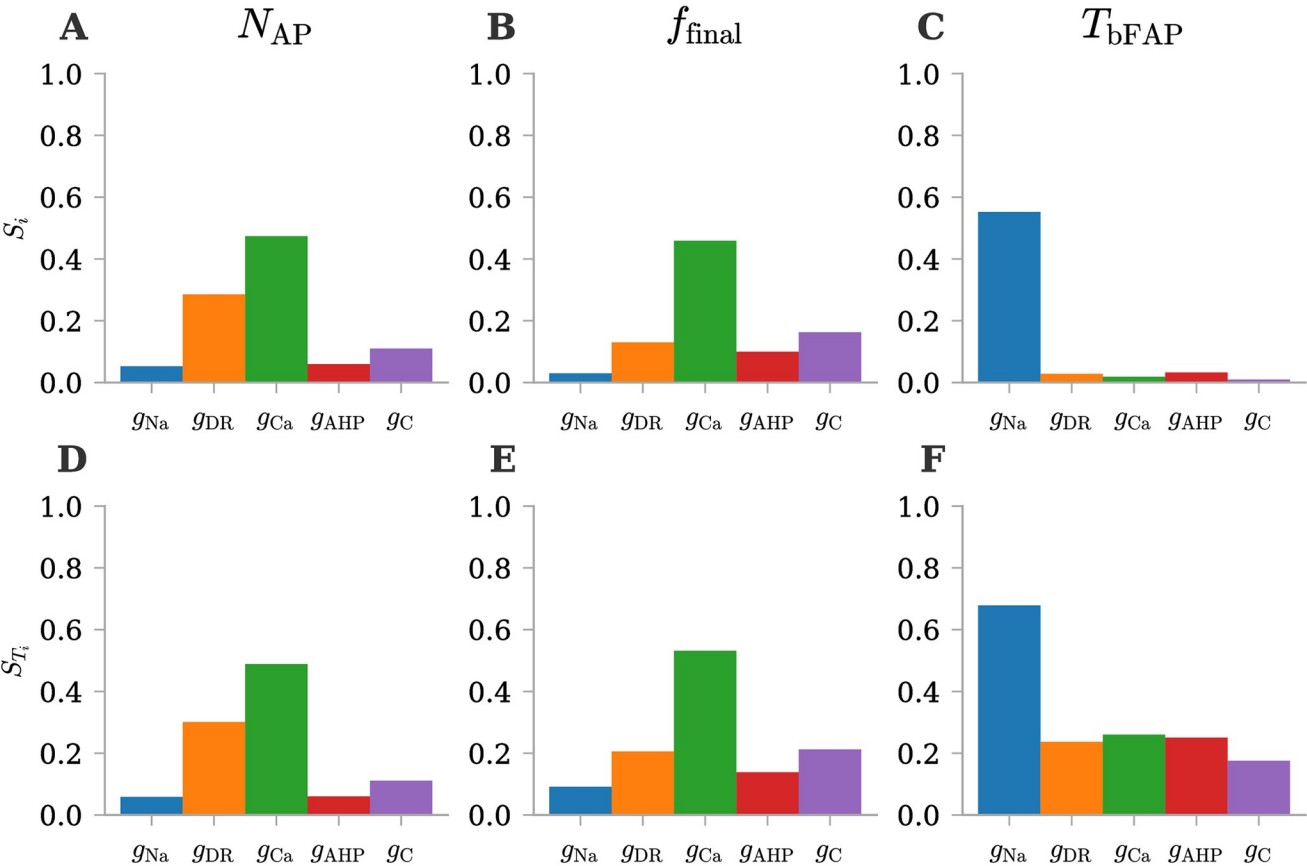

**Fig 4. First and Total order Sobol' indices for $\phi_{msn}$ QoIs under physiological conditions.** Sensitivity analysis with Sobol' indices for three QoIs (number of action potentials, $N_{AP}$, final firing frequency, $f_{final}$, and time before the first action potential, $T_{bFAP}$). **(A-C)** First order Sobol' indices for selected QoIs. **(D-F)** Total order Sobol' indices for selected QoIs.

conductances, indeed, exhibits the most significant First Order and Total Order Sobol' indices. Consequently, it becomes apparent that the variance in $N_{AP}$ and $f_{final}$ is predominantly driven by these three conductances ($g_{DR}$, $g_C$, and $g_C$). In contrast, an examination of the First Order Sobol' indices for $T_{bFAP}$ implies that the input parameter $g_{Na}$ owns a greater influence on output variance compared to the other input parameters (Fig 4C). However, this assertion does not hold as strongly when examining the Total Order indices. While $g_{Na}$ retains a substantial index value, all the other conductances also contribute significantly to the higher order effects through their interactions (Fig 4F).

**Comparing different versions of time-dependant Sobol' indices helps unravel the complex dynamics of $[K^+]_{se}$.** To comprehensively address the uncertainties and sensitivities inherent to a slow dynamic variable such as $[K^+]_{se}$, we conducted an in-depth analysis over the course of time (Fig 5). Fig 5A illustrates how the model's uncertainty evolves throughout the simulation. This uncertainty starts to grow at 0.2 s with the initiation of the stimulus current and continues to expand as firing activity progresses, resulting in increasingly wide prediction intervals. This highlights the importance of performing an accurate sensitivity analysis that takes into account this variance fluctuation. Indeed, we computed the Total Order Sobol' indices over time using three distinct approaches: the standard index $S_{T_i}$ (Fig 5B), the weighted index $S_{T_i}^W$ (Fig 5C), and the generalized version $\mathfrak{S}_{T_i}$ (Fig 5D). It is challenging to interpret standard indices due to the variability of the output over time and the oscillations within the indices themselves. Conversely, their generalized counterparts offer a more robust and meaningful perspective. Delving into the latter, we observe how initially, prior to firing activity, the sole parameter influencing the output is $g_{DR}$, with a Sobol' index nearly reaching 1. As firing begins, all parameter indices display a peak followed by a decay, eventually stabilizing over time. Among these, $g_{DR}$ and $g_C$ emerge as the key contributors to output uncertainty, with their Total Order Sobol' indices reaching 0.3 at the simulation's end. In contrast, $g_{Na}$ and $g_C$ exhibit indices around 0.04. Notably, $g_{AHP}$ initially experiences a peak around 0.6 s, followed by a brief dip; however, as time progresses, its index surpasses that of $g_{DR}$ and $g_C$, reaching a final value of 0.42. While the standard Sobol' indices and their generalized versions might ignore the initial low variance and prematurely highlight the significance of $g_{DR}$, weighted Sobol' indices provides a clearer contrast. Weighted Sobol' indices offer improved understanding, particularly at the simulation's onset and conclusion—instances when dynamic changes occur due to the activation and deactivation of the stimulus current. Indeed, within the interval (0, 0.2) s, no input parameter affects output uncertainty due to minimal variance. During firing activity, despite oscillations complicating interpretation, the two parameter groups already discussed emerge: one exerting greater influence ($g_{DR}$ and $g_C$) and another with lesser impact ($g_{Na}$ and $g_C$), while $g_{AHP}$'s importance increases to eventually surpass the others. When firing ceases, it is evident how the significance of $g_{DR}$ and $g_C$ diminishes immediately. This highlights that these parameters only actively influence the output during firing. Conversely, the other parameters, when the oscillatory dynamics ceased, display an almost constant value, preserving their relative importance on the output uncertainty.

## Uncertainty quantification and sensitivity analysis under pathological conditions

**Input uncertainty significantly influence the time of the start of the depolariziation block.** In this section, we address the outcomes of our analysis regarding the uncertainty of the state variable $\phi_{msn}$ under pathological conditions. Specifically, in this second test case, the selected QoIs were the time of the start of the depolariziation block ($T_{sDP}$) and the time before the first action potential ($T_{bFAP}$). We executed the identical procedure outlined in the Results

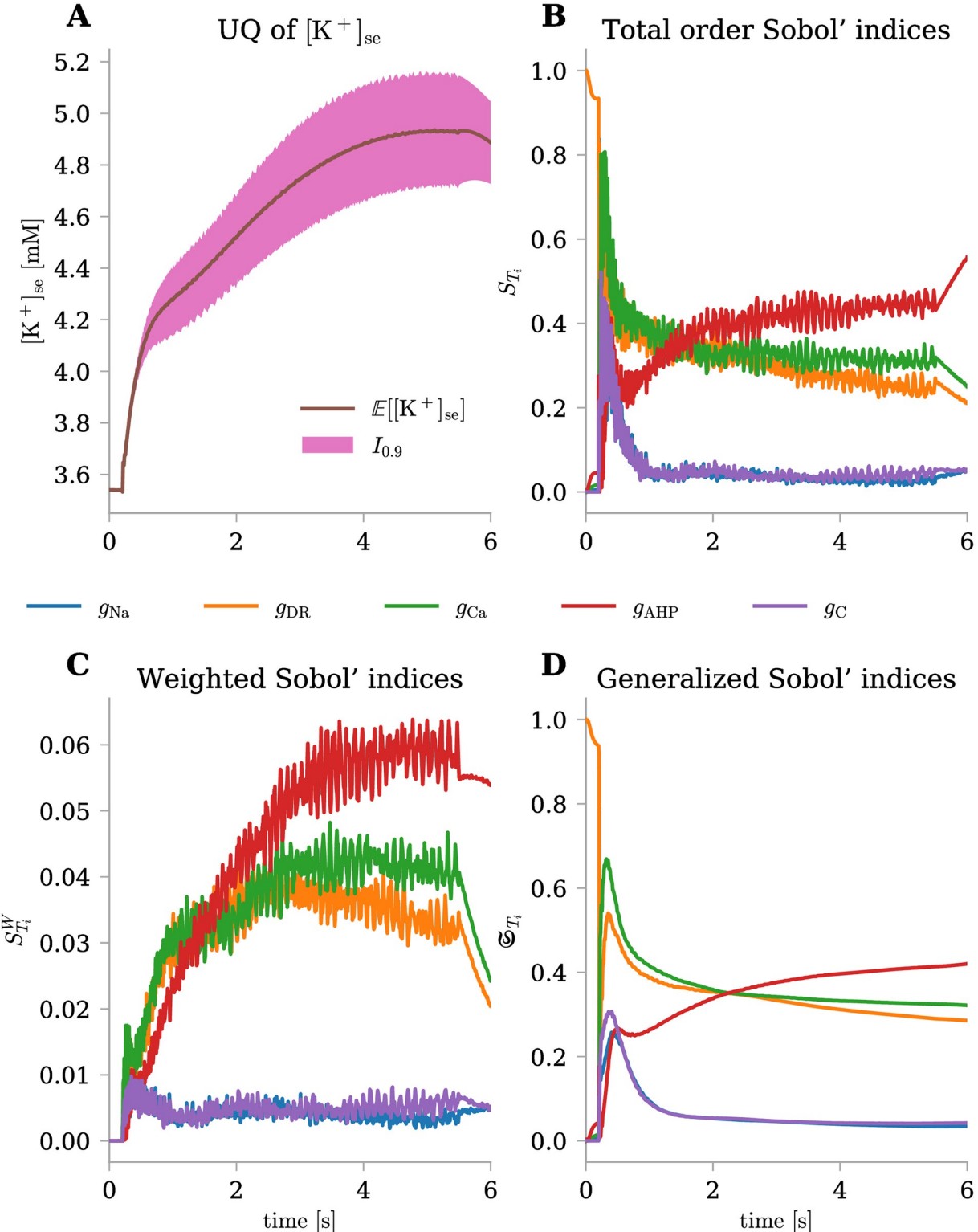

**Fig 5. Uncertainty quantification and sensitivity analysis for $[K^+]_{se}(t)$ under physiological conditions.** The uncertainty level on input parameters was fixed at $\hat{\sigma} = 5\%$. **(A)** Mean ($\mathbb{E}$) and 90% prediction interval ($I_{0.9}$) for $[K^+]_{se}(t)$ evaluated at each time-step. **(B-D)** Total order Sobol' indices ($S_{T_i}$), weighted Total order Sobol' indices ($S_{T_i}^W$), and generalized Total order Sobol' indices ($\mathfrak{S}_{T_i}$) over time for the five uncertain parameters (different colors).

section for physiological conditions. Initially, we set the uncertainty at a moderate level of $\hat{\sigma} = 5\%$ and generated histograms (Fig 6A and 6B) portraying approximate probability density functions for the selected QoIs. The results revealed that $T_{\text{sDP}}$ spans approximately from 3.25 to 4.25 s, while $T_{\text{bFAP}}$ falls within the range of 3.5 to 3.8 ms. Notably, the variability in $T_{\text{sDP}}$ is

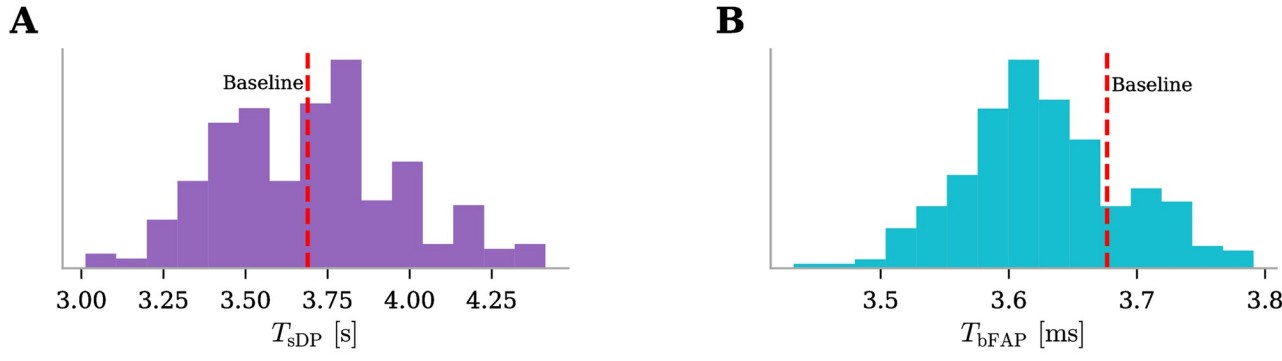

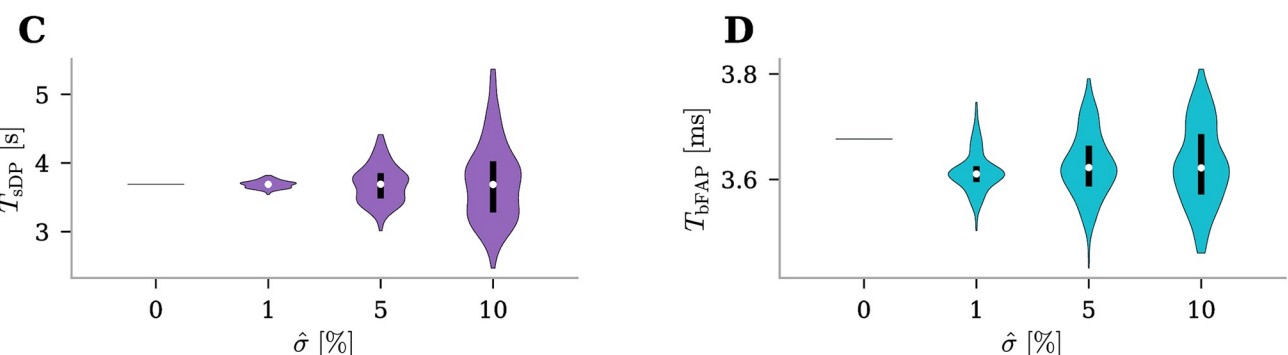

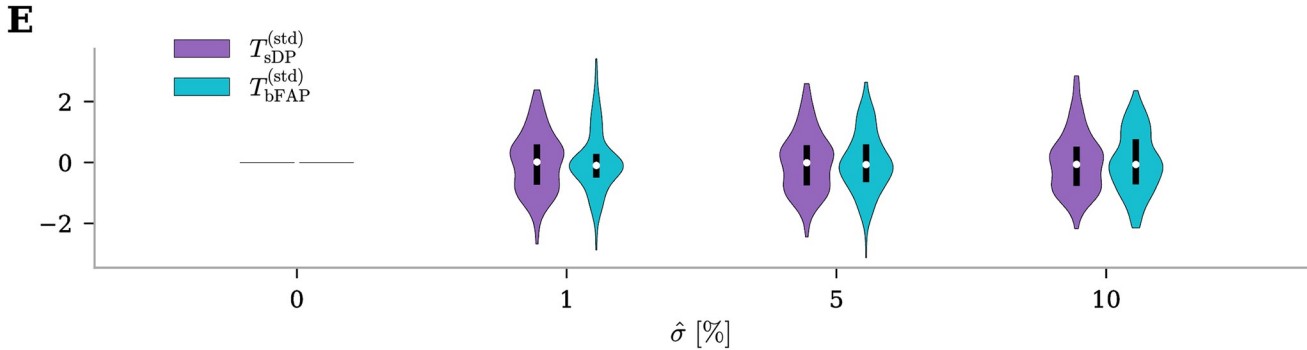

**Fig 6. Uncertainty quantification for $\phi_{\text{msn}}$ QoIs under pathological conditions.** Outputs for two QoIs (time of the start of the depolarization block, $T_{\text{sDP}}$, and time before the first action potential, $T_{\text{bFAP}}$). **(A-C)** Histograms of QoIs with moderate uncertainty level on input parameters, $\hat{\sigma} = 5\%$. The red dashed lines indicate the values of QoIs when the edNEG model is evaluated using baseline parameter values. **(D-F)** Violin plot of QoIs with increasing level of uncertainty in the input parameters, $\hat{\sigma} = \{0, 1, 5, 10\}\%$ (x-axis). The width of each violin at any given point indicates the density of the QoI at the corresponding uncertainty level. The white dots represent the median and the black bold lines the data that lie within quartiles. **(G)** Grouped violin plot of standardized QoIs ((data—mean)/standard deviation) with increasing level of uncertainty in the input parameters.

substantial, encompassing a difference of around 1 s, despite a clustering of values around the baseline. In contrast, the range of $T_{\mathrm{bFAP}}$ is slightly narrower, approximately 0.3 ms, and both QoIs exhibit a Gaussian-like distribution. Subsequently, we changed the uncertainty levels, varying $\hat{\sigma} \in \{0, 1, 5, 10\}$ (Fig 6C–6E). As observed earlier, $T_{\mathrm{sDP}}$ is markedly influenced by input uncertainty. With increasing uncertainty, the distributions widen, transitioning from a narrow range around 3.7 s at $\hat{\sigma} = 1\%$ to spanning between 2.5 and 5.0 s at $\hat{\sigma} = 10\%$ (Fig 6C). Furthermore, $T_{\mathrm{bFAP}}$ exhibits a moderate amplification in output uncertainty, with a broader distribution as uncertainty increases, although less pronounced than the effect on $T_{\mathrm{sDP}}$ (Fig 6D). Finally, in Fig 6E, we present standardized QoI distributions, categorized by distinct uncertainty levels. It is now evident that all QoIs are influenced by input uncertainty at comparable levels.

**The selected QoIs are influenced in distinct ways by the five uncertain parameters.** Following the same methodology applied in the physiological scenario, we conducted a sensitivity analysis using First and Total Order Sobol' indices (Fig 7). A notable observation emerges when comparing the first and total effects: the predominant source of uncertainty for $T_{\mathrm{sDP}}$ stems from first-order effects, given the negligible difference between their First and Total Order indices. In contrast, regarding $T_{\mathrm{bFAP}}$, a substantial difference is evident, indicating that its uncertainty is primarily attributed to interactions among input parameters. The results for $T_{\mathrm{sDP}}$ (Fig 7A and 7C) suggest that this QoI is highly sensitive to the input parameter $g_{\mathrm{DR}}$ since it exhibits the most significant First Order and Total Order Sobol' indices. On the other hand, an analysis of the First Order Sobol' indices for $T_{\mathrm{bFAP}}$ implies that the input parameter $g_{\mathrm{Na}}$ exerts a greater influence on output variance compared to the other input parameters (Fig 7B). However, when examining the Total Order indices, it becomes evident that all the other conductances play a substantial role in contributing to higher-order effects through their interactions (Fig 7D).

**The dynamics of $[\mathrm{K}^+]_{\mathrm{se}}$ exhibits sensitivity to different conductances at distinct stages.** Analogously to the analysis described in the section related to Fig 5, which refers to the slow dynamic variable $[\mathrm{K}^+]_{\mathrm{se}}$ in the physiological case, we conducted a comprehensive analysis over time under pathological conditions as well (Fig 8). Fig 8A illustrates the evolution of the model's uncertainty throughout the simulation. At the initial stages of the simulation, uncertainty is almost negligible, even after the stimulus current initiates at 0.2 s. However, uncertainty begins to increase around 3.5 s when the neuron enters the depolarization block and firing ceases. Eventually, when the stimulus current is turned off at 5.5 s, the variance widens significantly. Subsequently, we computed the Total Order Sobol' indices using three distinct approaches presented earlier: the standard index $S_{T_i}$ (Fig 8B), the weighted index $S_{T_i}^{W}$ (Fig 8C), and the generalized version $\mathfrak{S}_{T_i}$ (Fig 8D). Fig 8B and 8D shows that before firing activity, the sole parameter influencing the output is $g_{\mathrm{DR}}$ with a Sobol' index nearly reaching 1. As firing begins, all parameter indices, except $g_{\mathrm{Na}}$, display a peak followed by a decline, with $g_{\mathrm{AHP}}$ showing a slower but similar dynamics. Conversely, $g_{\mathrm{Na}}$ appears to be the parameter most affecting the system's dynamics during firing activity, surpassing all others and reaching a Sobol' index around 0.6. As the system enters the depolarization block, a different situation emerges. While the importance of $g_{\mathrm{Na}}$ decreases strongly, the contribution of $g_{\mathrm{DR}}$ drastically increases, eventually surpassing that of $g_{\mathrm{Na}}$, revealing its importance especially at the beginning of the depolarization block. Since $\mathfrak{S}_{T_i}$ also considers the temporal impact on Sobol' indices, Fig 8D demonstrates that $g_{\mathrm{DR}}$ not only strongly influences the initial phase of the depolarization block, but that this influence persists throughout the dynamics as time progresses until the simulation ends. While standard Sobol' indices and their generalized version might overlook the initial low variance and prematurely highlight the significance of $g_{\mathrm{Na}}$, Fig 8C provides a

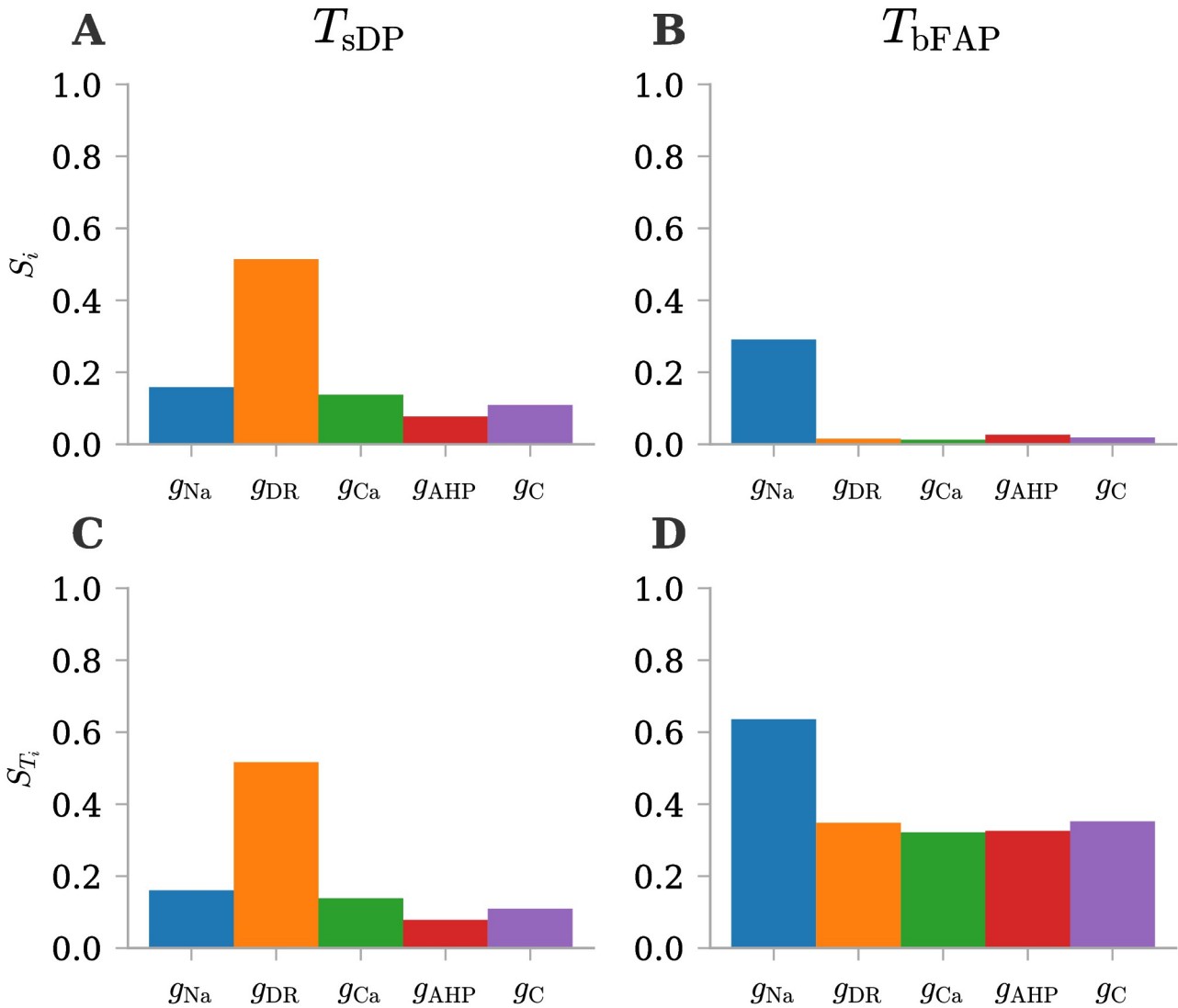

**Fig 7. First and Total order Sobol' indices for $\phi_{\text{msn}}$ QoIs under pathological conditions.** Sensitivity analysis with Sobol' indices for two QoIs (time of the start of the depolarization block, $T_{\text{sDP}}$, and time before the first action potential, $T_{\text{bFAP}}$). **(A-C)** First order Sobol' indices for selected QoIs. **(D-F)** Total order Sobol' indices for selected QoIs.

clearer contrast. Weighted Sobol' indices take into account the considerably lower variance at the beginning of our simulation compared to the second part when the system enters the depolarization block. We can still observe that at the beginning, $g_{\text{Na}}$ has the most significant impact on the dynamics, but the importance of $g_{\text{DR}}$ during the depolarization block is much higher and strongly influences the dynamics, as the variance of the output variable is significantly higher. Finally, when firing ceases, it becomes evident that the significance of all input parameters increases immediately due to the strongly increased variance.

## Discussion

In this paper, we introduce effective strategies for conducting UQ and GSA on neuron models incorporating ion concentration dynamics. Specifically, we study the edNEG model from

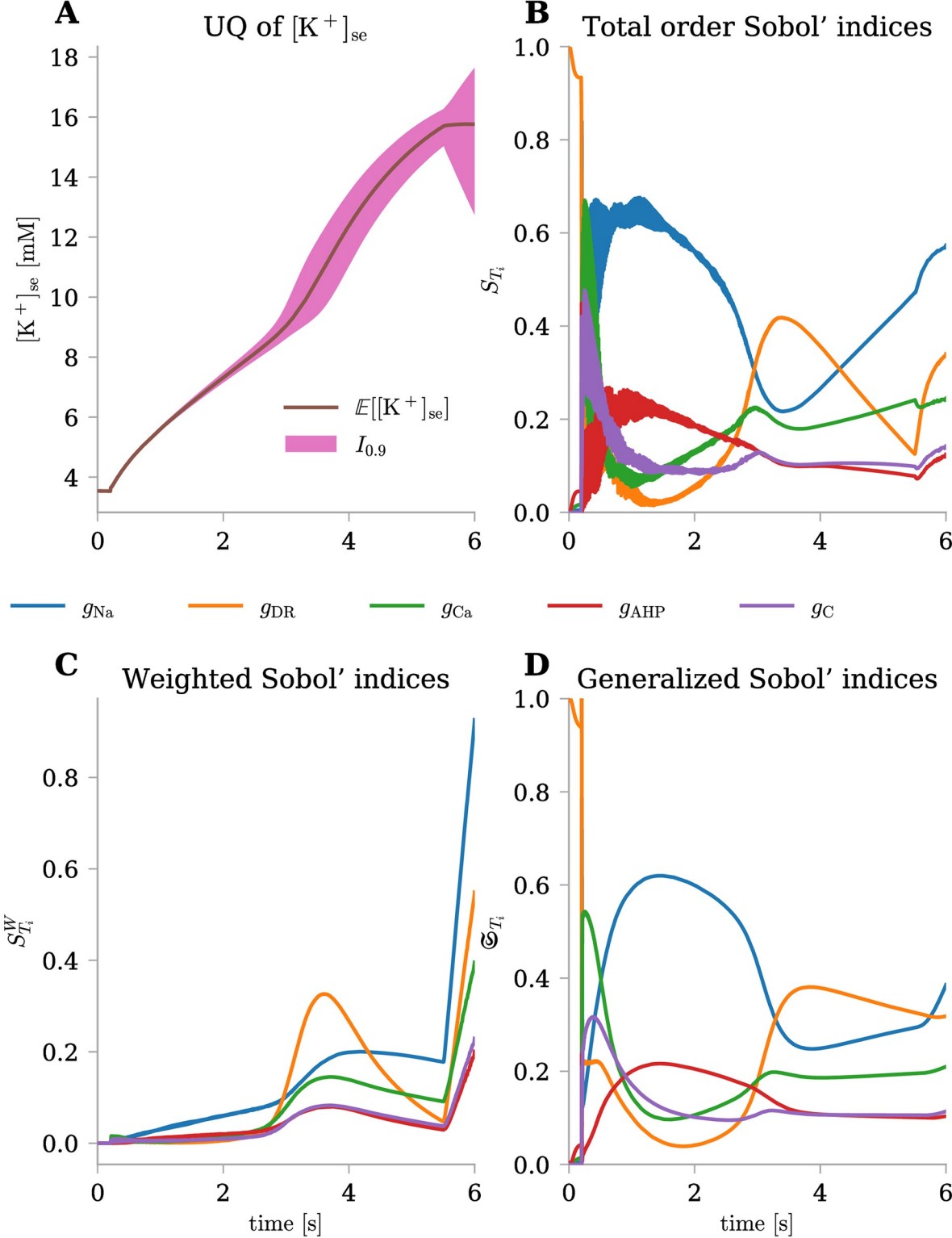

**Fig 8. Uncertainty quantification and sensitivity analysis for $[K^+]_{se}(t)$ under pathological conditions.** The uncertainty level on input parameters was fixed at $\hat{\sigma} = 5\%$. **(A)** Mean ($\mathbb{E}$) and 90% prediction interval ($I_{0.9}$) for $[K^+]_{se}(t)$ evaluated at each time-step. **(B-D)** Total order Sobol' indices ($S_{T_i}$), weighted Total order Sobol' indices ($S_{T_i}^W$), and generalized Total order Sobol' indices ($\mathfrak{S}_{T_i}$) over time for the five uncertain parameters (different colors).

Sætra et al. 2021 [20]. The edNEG model predicts ion concentrations, electrical potentials, and volumes within a six-compartmental system representing a neuron, ECS, and glial cells, formulated as a system of 34 ODEs. Our approach involved three key aspects:

1. Firstly, by considering effective solving strategies, we achieved simulations that were 15 times more time-efficient than our initial implementation from Sætra et al. 2021 [20], enabling an in-depth analysis of the model's dynamics.

2. Secondly, we implemented a factor-fixing analysis to isolate the parameters directly influencing the dynamical state of the system. This analysis narrowed our focus to the five membrane conductances listed in Table 2.

3. Thirdly, we did a careful selection of which QoIs to study. The fast dynamics exhibited by $\phi_{\mathrm{msn}}$ necessitated the selection of specific spiking features, a common consideration in neuroscience [22]. Conversely, the slower dynamics associated with the ECS potassium ion concentration, $[\mathrm{K}^+]_{\mathrm{se}}$, allowed for a time-dependent examination. This decision played a vital role in ensuring meaningful results, especially when dealing with multiple dynamic patterns arising from the model.

In our study, our emphasis was on understanding the role of uncertain parameters during neuronal firing. Our in-depth analysis identified specific parameters influencing chosen spiking features and the dynamics of $[\mathrm{K}^+]_{\mathrm{se}}$ at different simulation time intervals. Further elaboration on these findings will be provided in the following subsections, culminating in a discussion on future perspectives.

## Understanding the source of uncertainty gives insights into the model's underlying dynamics

In our analysis of $\phi_{\mathrm{msn}}$ under physiological conditions, our key findings can be summarized as follows: for $N_{\mathrm{AP}}$ and $f_{\mathrm{final}}$, most of the uncertainty arises from the first order effects of the conductances $g_{\mathrm{DR}}$, $g_{\mathrm{C}}$, and $g_{\mathrm{C}}$. It is worth noting that the calcium current contributes to the depolarization of the dendritic compartment, whereas the potassium delayed rectifier current serves to repolarize the somatic compartment, terminating the firing of action potentials. This underscores the importance of accurately estimating these conductances to mitigate uncertainties in the model predictions, as both $N_{\mathrm{AP}}$ and $f_{\mathrm{final}}$ stand out as profoundly impacted by input uncertainties. Conversely, for $T_{\mathrm{bFAP}}$, total effects dominate the output uncertainty, driven by all conductances, but especially by $g_{\mathrm{Na}}$. Indeed, since the sodium current depolarizes the somatic compartment to initiate firing of action potentials, it follows that variations in the sodium channel conductance, $g_{\mathrm{Na}}$, notably influence the time before the first action potential. This influence is most pronounced in its first order effects. In contrast, all other conductances affect this particular QoI only through interactions among themselves.

In the analysis of $[\mathrm{K}^+]_{\mathrm{se}}$ under physiological conditions, we explored three different implementations of the Total Order Sobol' indices for time-dependent processes. Our findings provide robust insights into how not only the input parameters but also time itself influence its dynamics. We noticed the emergence of two distinct parameter groups: one with a more pronounced impact, consisting of $g_{\mathrm{DR}}$ and $g_{\mathrm{C}}$, and another with a comparatively smaller influence, represented by $g_{\mathrm{Na}}$ and $g_{\mathrm{C}}$. Interestingly, the significance of $g_{\mathrm{AHP}}$ grew over time, eventually surpassing the other parameters in importance. Indeed, from a biological perspective, the voltage-dependent potassium after-hyperpolarization (AHP) current has a slow time constant, causing the cell to stay hyperpolarized [45] and giving its contribution as the simulation time progresses.

## Comparing how uncertainty affects both physiological and pathological activity unravels different scenarios

When focusing on the second test case under pathological conditions, we observe that the time of the start of the depolarization block, $T_{sDP}$, is substantially affected by input uncertainty. This highlights the profound impact of active ion channel conductances, especially $g_{DR}$, on the neuron's ability to tolerate a strong input current. Furthermore, the time before the first action potential, $T_{bFAP}$, is moderately influenced by input uncertainty, as in the physiological case. Specifically, the conductance $g_{Na}$ significantly contributes to first-order effects, while all other conductances exert a strong influence on $T_{bFAP}$ through higher-order interactions. Comparing varying levels of uncertainty on $T_{bFAP}$ reveals only a minimal increase in output uncertainty, both for the pathological and physiological scenario. This phenomenon may be attributed to $T_{bFAP}$ being a QoI measured at the onset of firing when the system variance is low, and time has not significantly contributed to overall uncertainty.

A notable scenario arises when comparing the analyses of the time-dependent output $[K^+]_{se}$. Under pathological conditions, the variance of $[K^+]_{se}$ remains minimal during the fast dynamics between 0.2 and 3.7 s, and increases as the neuron enters the depolarization block after around 3.8 s. This contrasts with physiological conditions, where the variance of $[K^+]_{se}$ starts to rise just after the onset of the stimulus current. This observation highlights the significance of the active ion channels' uncertainty during the depolarization block in pathological dynamics. Additionally, upon turning off the stimulus current at 5.5 s, there is a drastic increase in the variance of $[K^+]_{se}$. This phenomenon may arise from different dynamic scenarios resulting from varied parameter settings. However, given our study's focus on the dynamical state, we leave the investigation of this subject to future explorations.

When delving into the contributions of individual parameters to the uncertainty in $[K^+]_{se}$, notable differences emerge compared to physiological conditions. While the physiological case emphasizes the importance of $g_{DR}$ and $g_C$ during firing activity, pathological conditions are much more impacted by the uncertainty in $g_{Na}$, even though the overall variance during this stage is minimal. Conversely, the influence of $g_{DR}$ only begins to rise as the neuron enters the depolarization block. This once again underscores the impact of this input parameter under pathological conditions, particularly during the depolarization block.

## Future perspectives

In conclusion, we would like to present some potential avenues for future research in sensitivity analysis.

Firstly, while this paper delved into the influence of five selected parameters, the model's extensive parameter space allows for alternative choices. It would be valuable, for example, to explore the influence of "non-dynamic" parameters, such as leak conductances and cotransporters, on both system dynamics and volume changes during extended simulation periods. We anticipate this parameter group to be somewhat more challenging to analyze, as they affect the system dynamics through alterations of the resting state. Indeed, this presents an additional challenge in comprehending how uncertainty influences the system and would be an interesting topic for research in the near future.

Secondly, when selecting probability distributions for the input parameters, our approach involved setting all distributions to uniform, without delving into how they are determined. Moving forward, there is potential to explore empirically-derived distributions.

Thirdly, the analysis conducted here focused on just two model outputs, $\phi_{msn}$ and $[K^+]_{se}$, despite the model containing 34 ODEs. Exploring other outputs within the framework of GSA, such as the volumes of the compartments, could provide further insights. Given the slow

time scale of volume changes, one could adopt a similar approach as the one we utilized for $[K^+]_{se}$.

Additionally, our GSA concentrated on just a few QoIs for $\phi_{msn}$. Considering the diversity of features available, a more extensive exploration of QoIs could enhance the understanding of uncertainty and sensitivity analysis. For instance, the `Uncertainpy` library [22] offers a wide array of QoIs that can be considered.

Lastly, the model's outcomes are influenced by the stimulus strength. Investigating the extent to which changes in the system's equilibrium depend on input parameter uncertainty could be an interesting future exploration. In this context, a potential development might involve conducting a bifurcation analysis, similar to the approach in Ghori and Kang 2023 [23], with the value of the stimulus current, $I_{stim}$, serving as a tunable parameter.

In summary, this study marks the first in-depth exploration of uncertainty assessment for neuron models incorporating ion concentration dynamics. Our sensitivity analysis sheds light on the critical parameters influencing our model's outputs and their interdependencies, and identifies the ones that require accurate estimation to reduce uncertainty in the results. Additionally, it provides valuable insights on the underlying biological mechanisms governing these dynamics under different conditions. We envision that the methodologies presented herein offer valuable guidelines for future studies, ultimately extending the application of neuron models with ion concentration dynamics.

## Supporting information

**S1 Appendix. Model parameters and initial conditions.**
(PDF)

## Acknowledgments

The authors acknowledge that parts of this paper have been published previously as a Master's thesis [46].

## Author Contributions

**Conceptualization:** Marte J. Sætra.

**Formal analysis:** Letizia Signorelli.

**Investigation:** Letizia Signorelli.

**Methodology:** Letizia Signorelli, Andrea Manzoni, Marte J. Sætra.

**Software:** Letizia Signorelli.

**Supervision:** Andrea Manzoni, Marte J. Sætra.

**Validation:** Letizia Signorelli, Andrea Manzoni, Marte J. Sætra.

**Visualization:** Letizia Signorelli, Andrea Manzoni, Marte J. Sætra.

**Writing – original draft:** Letizia Signorelli, Marte J. Sætra.

**Writing – review & editing:** Andrea Manzoni.

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
