## [Decision Letter · Decision Letter 0]

4 Feb 2024

PONE-D-24-00537Uncertainty quantification and sensitivity analysis of neuron models with ion concentration dynamicsPLOS ONE

Dear Dr. Signorelli,

Thank you for submitting your manuscript to PLOS ONE. After careful consideration, we feel that it has merit but does not fully meet PLOS ONE’s publication criteria as it currently stands. Therefore, we invite you to submit a revised version of the manuscript that addresses the points raised during the review process.

We look forward to receiving your revised manuscript.

Kind regards,

Alexey Kuznetsov

Academic Editor

PLOS ONE

Journal Requirements:

**Additional Editor Comments:**

First, reviewers have raised concerns on model calibration. Therefore, the choice of parameter values requires better justification.

Second, reviewers found multiple typos and convoluted spots. Therefore, the manuscript is required to be corrected and carefully proofread by a proficient English speaker.

Additionally, the manuscript would improve if the discussion was written for a more general audience.

Reviewers' comments:

Reviewer's Responses to Questions

**Comments to the Author**

1. Is the manuscript technically sound, and do the data support the conclusions?

Reviewer #1: Yes

Reviewer #2: Yes

2. Has the statistical analysis been performed appropriately and rigorously? 

Reviewer #1: Yes

Reviewer #2: Yes

3. Have the authors made all data underlying the findings in their manuscript fully available?

Reviewer #1: Yes

Reviewer #2: Yes

4. Is the manuscript presented in an intelligible fashion and written in standard English?

Reviewer #1: Yes

Reviewer #2: Yes

5. Review Comments to the Author

Reviewer #1: This paper carried out uncertainty quantification analysis for a six-compartment neuron model incorporating ion concentration dynamics from a comprehensive viewpoint. The topic is interesting and important, and the English language is clear. I think the paper can be accepted after a minor revision.

The following two points should be improved.

I. I am wondering how the dynamical parameters and the non-dynamical parameters are taken in Table 1 and Table 2. The authors should add necessary demonstration or citation.

II. The authors should reexamine their whole context to eliminate the possibles errors. For example, in the abstract, the sentence "To mitigate computational cost, we employ surrogate modeling techniques, optimized using efficient numerical integration techniques." contains grammatical error due to "optimized using efficient numerical integration techniques". For another example, the sentence "Our sensitivity analysis not only sheds some light on the critical parameters influencing our model’s outputs, their interdependencies, and which ones demand precise estimation to mitigate uncertainty in the results" seems strange due to " and which.....", please see Line 638-640.

Reviewer #2: This paper investigates the effect of uncertain parameters on an electrodiffusive neuron-extracellular-gli (edNEG) neuron model through the use of uncertainty quantification (UQ) and sensitivity analysis (SA). The authors focus on the effect of uncertainties in a selected set of dynamic model parameters and examined the effects on the model under two different conditions, physiological and pathological. This paper discusses several common challenges for uncertainty quantification of neuron models with ion concentration dynamics and is a good showcase for how to approach these challenges within the field of neuroscience.

The main strength of the paper is the case study and discussion of the encountered challenges. This serves as a showcase for how common challenges related to UQ and SA of neuron models may be addressed, contributing to wider adoption of uncertainty quantification within neuroscience.

There are a few minor issues that preferably could be addressed to improve the article, but no major issues that fundamentally affect the work and conclusions.

The main issue that would strengthen the paper if it was improved is to generalize the discussions related to how the different challenges were solved, making it easier for others to apply them, as the authors mention they want the article to offer guidelines for others to follow.

One example is the section on “Numerical implementation and validation” of the edNEG model (line 165), which details how the edNEG model was optimized in order to perform UQ and SA. The specific optimizations needed are of course highly dependent on the specific model and could be the topic of multiple focused papers, however it would be useful to have more information why these specific changes were made and the thought process behind them, as that will help guide others. It would also be useful if the text could mention some general considerations that others can make. For example, why was the choice of a timestep of 10ms selected? What other timesteps did the authors consider to use and why were they discarded?

The same applies elsewhere, it would for example be useful to know why the selected quantities of interest were chosen for examination in the physiological and pathological cases (line 298-305). If the reader is provided with insight as to why these quantities of interest were the most relevant for this case study, it may be easier for them to make similar considerations for their own use case. There might also be other places where broader perspectives could enhance the article, perhaps for the Factor fixing section (line 231)?

Specific comments:

1. The first sentence of the abstract makes me interpret the paper to focus on inventing new computational or algorithmic approaches for more efficient UQ and SA. It would be beneficial with a rephrasing that shows the main strength of the paper, namely as a case study for how common challenges with UQ and SA in neuroscience can be approached and solved.

2. 53-54: “ To fully exploit the potential of surrogate modeling, it is essential to adopt efficient numerical integration techniques”. Would this not be even more essential if other methods than surrogate modeling were used, for example quasi-Monte Carlo methods?

3. 82-83: “Additionally, evaluating sensitivity indices for a time-dependent output (i.e., one for each time point) is computationally demanding,”. I am just curious, is this computationally demanding when compared to the computational cost of running the model evaluations?

4. 85-86: “we introduce a comprehensive and computationally efficient approach for conducting UQ and GSA on neuron models with ion concentration dynamic”. This sentence gives the same impression as in the abstract (1.). Perhaps rephrasing it would make the content of the paper more clear?

5. 99 - 100: “This efficiency demonstrates the feasibility of conducting sensitivity analysis on complex neuroscience models”. It would be beneficial to know the runtime for UQ and SA of the model with and without the performance improvements, since most readers likely do not know the performance of the unoptimized model and subsequently if the (rather impressive) speedup is necessary.

6. 178-179: The code is readable and freely available with instructions for how to run it, which is very good! My one suggestion would be to also archive the code on zenodo (https://zenodo.org/). That would give the code a DOI to cite and track it with and provide redundancy in case github repositories get deleted.

7. 248: Was there a reason an uncertainty of 15% was used in the factor fixing analysis, while for examining the dynamic parameters {0, 1, 5, 10} was used?

8. 258: “these variables were carefully selected”: It would be nice with a reference to where in the paper this ends up being expanded upon (see comment on main issue).

9. 261-266: “Implementation details”. To my understanding, this calculates the Sobol indices using quasi-Monte Carlo methods? Why are not surrogate models used, as that is mentioned earlier to be imperative for computational efficiency? It would also make the text more clear if the authors mention that the goal is to calculate the Sobol indices.

10. 319-320: “Outputs were generated by drawing a Monte Carlo sample of size 104 from the parameter distribution”. To my understanding, this sentence refers to how many samples were used for finding the 5th and 95th percentiles? (the nr_pc_mc_samplest argument?). If so, it might be more clear to say 5th and 95th percentiles instead of outputs, and I would put it after the next sentence “To enhance computational efficiency,” as that is the more important information in this context.

11. 323: “Rosenblatt transformation for dependent parameter”, the Rosenblatt transformation is not necessary unless there are dependencies in the input parameters, which seems to not be the case in the manuscript?

12. 350: It is a bit unclear to me how the total order Sobol indices for each group was found in the factor fixing analysis, is it the sum of the Sobol indices for all uncertain parameters in each group?

13. Fig 3, Fig 5, Fig 6, and Fig 8 could all benefit from having subtitles added to the subplots. The plots are readable, but that would make it easier and faster to interpret them.

6. PLOS authors have the option to publish the peer review history of their article (what does this mean?). If published, this will include your full peer review and any attached files.

Reviewer #1: No

Reviewer #2: **Yes: **Simen Tennøe

---

## [Author Response · Author response to Decision Letter 0]

21 Mar 2024

Dear Editor,

We hereby resubmit our manuscript “Uncertainty quantification and sensitivity analysis of neuron models with ion concentration dynamics” by Letizia Signorelli, Andrea Manzoni, and Marte J. Sætra for publication in PLOS ONE.

We thank the reviewers for their feedback and constructive comments.

Our detailed response is given in the attachment labeled Response to Reviewers.

---

## [Decision Letter · Decision Letter 1]

2 May 2024

Uncertainty quantification and sensitivity analysis of neuron models with ion concentration dynamics

PONE-D-24-00537R1

Dear Dr. Signorelli,

We’re pleased to inform you that your manuscript has been judged scientifically suitable for publication and will be formally accepted for publication once it meets all outstanding technical requirements.

Kind regards,

Alexey Kuznetsov

Academic Editor

PLOS ONE

Additional Editor Comments (optional):

Reviewers' comments:

Reviewer's Responses to Questions

**Comments to the Author**

1. If the authors have adequately addressed your comments raised in a previous round of review and you feel that this manuscript is now acceptable for publication, you may indicate that here to bypass the “Comments to the Author” section, enter your conflict of interest statement in the “Confidential to Editor” section, and submit your "Accept" recommendation.

Reviewer #2: All comments have been addressed

2. Is the manuscript technically sound, and do the data support the conclusions?

Reviewer #2: Yes

3. Has the statistical analysis been performed appropriately and rigorously? 

Reviewer #2: Yes

4. Have the authors made all data underlying the findings in their manuscript fully available?

Reviewer #2: Yes

5. Is the manuscript presented in an intelligible fashion and written in standard English?

Reviewer #2: Yes

6. Review Comments to the Author

Reviewer #2: All my comments have been addressed and I am happy to recommend the paper being accepted. The authors have done a solid job.

7. PLOS authors have the option to publish the peer review history of their article (what does this mean?). If published, this will include your full peer review and any attached files.

Reviewer #2: **Yes: **Simen Tennøe

---

## [Editor Report · Acceptance letter]

7 May 2024

PONE-D-24-00537R1 

PLOS ONE

Dear Dr. Signorelli, 

I'm pleased to inform you that your manuscript has been deemed suitable for publication in PLOS ONE. Congratulations! Your manuscript is now being handed over to our production team.

Kind regards, 

on behalf of

Dr. Alexey Kuznetsov 

Academic Editor

PLOS ONE